# A Phase Difference Measurement Method for Integrated Optical Interferometric Imagers

Jialiang Chen [1,2]⬤, Qinghua Yu [1,*], Ben Ge [1,2], Chuang Zhang [1,2], Yan He [1,2] and Shengli Sun [1]

[1]   Key Laboratory of Intelligent Infrared Perception, Chinese Academy of Sciences, Shanghai Institute of Technical Physics of Chinese Academy of Sciences, Shanghai 200083, China
[2]   University of Chinese Academy of Sciences, Beijing 100049, China
*   Correspondence: yuqinghua@mail.sitp.ac.cn

**Abstract:** Interferometric imagers based on integrated optics have the advantages of miniaturization and low cost compared with traditional telescope imaging systems and are expected to be applied in the field of space target detection. Phase measurement of the complex coherence factor is crucial for the image reconstruction of interferometric imaging technology. This study discovers the effect of the phase of the complex coherence factor on the extrema of the interference fringes in the interferometric imager and proposes a method for calculating the phase difference of the complex coherence factor of two interference signals by comparing the extrema of the interferometric fringes in the area of approximate linear change in the envelope shape to obtain the phase information required for imaging. Experiments using two interferometric signals with a phase difference of $\pi$ were conducted to verify the validity and feasibility of the phase difference measurement method. Compared with the existing phase measurement methods, this method does not need to calibrate the position of the zero optical path difference and can be applied to the integrated optical interferometric imager using a single-mode fiber, which also allows the imager to work in a more flexible way. The theoretical phase measurement accuracy of this method is higher than 0.05 $\pi$, which meets the image reconstruction requirements.

**Keywords:** integrated optics; interferometric imaging; interference fringes; phase difference measurement

## 1. Introduction

High-resolution planet detection missions require the support of large-aperture telescopes, such as the Very Large Telescope, which obtained images of the newborn exoplanet PDS 70b [1,2], and optical infrared interferometers such as the Very Large Telescope Interferometer and the Center for High Angular Resolution Array for observing the formation of planetary systems [3]. However, for satellite payloads, both large-aperture conventional telescopes and long-baseline optical infrared space interferometers are much more expensive and technically difficult to manufacture [4,5]. In 2013, in order to achieve the goal of high-resolution exploration of Jupiter, Lockheed Martin introduced the concept of the segmented planar imaging detector for electro-optical reconnaissance (SPIDER) [6]. This new imaging detector, based on the principle of interferometric imaging, takes full advantage of the integration and miniaturization of integrated optical devices, which can greatly reduce the volume, quality, and power consumption of the imaging system. Because freedom from atmospheric turbulence [7] in space will reduce the difficulty of phase measurement of interferometric imagers, the integrated optical interferometric imager represented by SPIDER is suitable as a payload for planetary detection. Currently, the imager is still under continuous development and is moving towards practical applications.

Current simulation studies on integrated optical interferometric imagers focus on how to improve the imaging performance of imagers, such as changing the sampling frequency by changing the form of lens array arrangement to improve the imaging performance of

imagers, including a hierarchical multistage sampling lenslet array arrangement [8,9], a checkerboard arrangement [10], a hexagonal arrangement [11], etc. There are also studies to optimize the imager design using compressive sensing algorithms [12,13] and to improve the reconstructed image quality by investigating improved image reconstruction algorithms for the problem of insufficient sampling frequency [14,15]. As for experiments, SPIDER has proposed a total of four generations of PIC designs [16–19], among which the third-generation design reconstructed images in the laboratory to verify the principle and feasibility of the integrated optical interferometric imager [18]. The fourth-generation design enables on-chip integration of detectors, CMOS (Complementary Metal Oxide Semiconductor) trans-impedance-amplifiers, and related imager devices on photonic integrated circuits (PICs), enabling on-chip detection of 0.1 μW optical signals and a $0-\pi$ phase shift range, greatly enhancing imager integration [19]. However, limited by the size of silicon wafers, the baseline length of current integrated optical interferometric imagers can only reach a maximum of about 40 cm, limiting the resolution of the imager [20]. This limitation can be easily broken through the connection of single-mode fibers [21–25]. However, the fiber optical path difference (OPD) drift caused by vibration and thermal environment will make the phase of the complex coherence factor difficult to measure [26].

Interferometric imagers require the modulus and phase of the complex coherence factor corresponding to the spatial spectrum of the imaged object to reconstruct the image. The spatial spectrum depends on the spacing between the subapertures and the distance from the light source, while the modulus and phase information are contained in the visibility and position information of the interference fringes generated by the corresponding baselines, respectively [27,28]. The common methods of fringe detection and measurement are divided into two categories [29]—one involves obtaining the complete coherence envelope by means of delay line scanning [30,31], while the other involves measuring at or near the apex of the coherence envelope, such as phase tracking and group delay tracking [32,33]. The modulus of the complex coherence factor can be easily measured by combining the above two methods, while the phase information is usually determined from the position of the fringes at zero OPD. In an astronomical interferometer, the channeled spectrum is commonly used, i.e., the broadband optical signal is dispersed and recorded with an imaging detector, and the location of zero OPD is determined by observing the number of fringes at different OPD [7,29]. However, this approach increases the complexity of integration of the integrated optical interferometric imager. In contrast, delay lines and phase shifters are easier to integrate in PIC, so it will be more convenient for the integrated optical interferometric imager to obtain phase information by scanning the interference fringes. However, it is difficult to directly obtain the exact position of the zero OPD using the scanned interference fringe alone. For example, the position of the center of the interference fringe can be estimated using the envelope fitting method [30,31], but the result is affected by the measurement accuracy of the extreme of the interference fringe and the shape of the envelope, making it difficult to accurately identify the position of the zero OPD. Reference [18] demonstrated in the laboratory the results of an integrated optical interferometric imager in obtaining the modulus and phase information of the complex coherence factor by scanning the interference fringes and reconstructing the image, but the experiment used a pre-calibrated zero OPD method. For an imager with an integrated 90° optical hybrid [6,34], when the zero OPD of the instrument is pre-calibrated, the phase of the complex coherence factor can be directly obtained by the balanced detectors and the readout circuit. However, when the external OPD of the instrument changes, such as changing the orientation of the entire imaging system, the actual zero OPD position will deviate from the pre-calibrated zero OPD, leading to errors in phase measurement, thus limiting the flexible use of the integrated optical interferometric imager [16]. If a single-mode fiber is used to extend the baseline length of the imager, the resulting drift in the OPD of the fiber will also lead to a change in the current zero OPD, which also leads to an error in the measured phase. Therefore, it is necessary to develop a method for

measuring the phase of the complex coherence factor without pre-calibration of zero OPD for integrated optical interferometric imagers.

In this paper, a method is proposed to calculate the phase difference of the complex coherence factor of two interference signals by comparing the extrema of the interference fringes in the area of approximate linear change in the envelope shape without calibrating the position of zero OPD in order to obtain the phase information required for imaging. This manuscript is organized as follows. In Section 2, based on the extended expression of the Van Cittert–Zernike theorem, the equation description of the complete interference fringes produced by the overlay of two narrow-band optical signals from an incoherent extended light source is derived, whereby the relationship between the corresponding adjacent extrema of the interference fringe with different phases and the phase difference of the interference signal is analyzed, and the method of calculating the phase difference by comparing the extrema of the normalized interference signal in the region of approximately linear variation of the envelope shape is proposed in combination with simulations. Section 3 presents the phase difference measurement results of two interferometric signals with theoretical phase difference $\pi$ under two experimental schemes—amplitude-division interference and wavefront-division interference. The experimental results verify the validity and feasibility of the phase difference measurement method, and the maximum error of the measured phase difference is $0.15\pi$. The phase measurement method not only provides new ideas for phase measurement for integrated optical interferometric imager but also has some reference significance for astronomical optical interferometry phase measurement. At the same time, since the experiments in this paper use a polarization-maintaining (PM) single-mode fiber, the experimental results can also provide a certain reference when extending the baseline length of the imager through the fiber and help to promote the application of integrated optical interferometric imagers.

## 2. Materials and Methods

### 2.1. Complete Representation of Interference Fringes

The schematic diagram of the integrated optical interference imaging system is shown in Figure 1. Light from the incoherent light source distributed in region $D$ is received by the lenslet array after propagation through air and other media. The optical signal is coupled by the lenslet into the PIC, and after beam splitting by the array waveguide grating and phase modulation by the phase shifter, the optical signal belonging to the same spectral channel interferes in the $2 \times 2$ coupler, and the interference signal will be recorded by the balanced detectors. The phase shifter changes the OPD to produce interference fringes, which contain information about the complex coherence factor of the spatial frequency corresponding to the baseline. Under quasi-monochromatic conditions, the luminance distribution of the light source $I(\alpha, \beta)$ [27] can be obtained from the inverse Fourier transform by measuring the complex coherence factor $|\mu|e^{i\varphi}$ corresponding to different baselines according to the Van Cittert–Zernike theorem, where $|\mu|$ is the modulus of the complex coherence factor and $\varphi$ is the phase of the complex coherence factor.

Based on the extension of the Van Cittert–Zernike theorem, the mutual coherence function of the optical signals received by the two endpoints $P_1$ and $P_2$ of the baseline on the surface of the lenslet array can be described by the following Equation [35]:

$$\Gamma(P_1, P_2, \tau) = \sqrt{I(P_1)}\sqrt{I(P_2)}\gamma(P_1, P_2, \tau) = \int_0^\infty e^{-2\pi i v \tau} dv \iint_D I(\alpha, \beta, v)\frac{e^{ik(R_1-R_2)}}{R_1 R_2} d\alpha d\beta \quad (1)$$

where

$$I(P_1) = \Gamma(P_1, P_1, 0) = \int_0^\infty dv \iint_D \frac{I(\alpha, \beta, v)}{R_1{}^2} d\alpha d\beta, \quad (2)$$

$$I(P_2) = \Gamma(P_2, P_2, 0) = \int_0^\infty dv \iint_D \frac{I(\alpha, \beta, v)}{R_2{}^2} d\alpha d\beta. \quad (3)$$

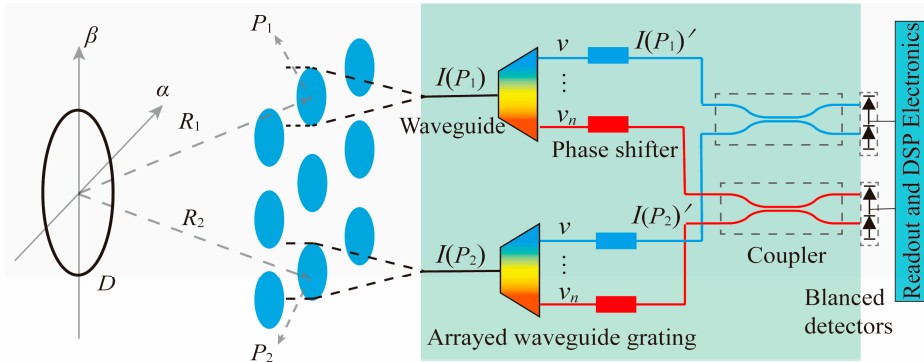

**Figure 1.** Schematic diagram of the integrated optical interferometric imaging system.

$I(P_1)$ and $I(P_2)$ are the light intensities of the light signals received at $P_1$ and $P_2$, respectively, $\gamma(P_1, P_2, \tau)$ is the complex coherence of the light signals at $P_1$ and $P_2$ at time delay $\tau$, $I(\alpha, \beta, v)$ is the light intensity per unit area of the extended light source in the frequency band $(v_0 - \frac{\Delta v}{2}, v_0 + \frac{\Delta v}{2})$, and $R_1$ and $R_2$ are the distances from the light source to $P_1$ and $P_2$, respectively.

From Equation (1), when $\tau = 0$,

$$\gamma(P_1, P_2, 0) = \frac{\int_0^\infty dv \iint_D I(\alpha, \beta, v) \frac{e^{ik(R_1 - R_2)}}{R_1 R_2} d\alpha d\beta}{\sqrt{I(P_1)} \sqrt{I(P_2)}} = \frac{\int_0^\infty dv \iint_D I(\alpha, \beta, v) e^{ik(R_1 - R_2)} d\alpha d\beta}{\int_0^\infty dv \iint_D I(\alpha, \beta, v) d\alpha d\beta} \quad (4)$$

Assuming that the intensity of the light source $I(\alpha, \beta, v)$ is the same for all frequencies in the same face of the light source region D in the narrow band $(v_0 - \frac{\Delta v}{2}, v_0 + \frac{\Delta v}{2})$, then $I(\alpha, \beta, v)$ can be considered as a constant independent of $v$. At this time,

$$\gamma(P_1, P_2, 0) = \mu(P_1, P_2) = |\mu| e^{i\varphi}. \quad (5)$$

Substituting Equation (5) into Equation (1) and taking into account the frequency response of the imaging system $T(v)$, the complex coherence degree of the optical signal at the coupler is obtained as

$$\gamma(P_1', P_2', \tau) = \mu(P_1, P_2) \frac{\int_0^\infty T(v) e^{-2\pi i v \tau} dv}{\int_0^\infty T(v) dv} = |\mu| \frac{\int_0^\infty T(v) e^{i\varphi - 2\pi i v \tau} dv}{\int_0^\infty T(v) dv}. \quad (6)$$

Assuming that the total dispersion of the light signal transimitted in all media is 0, then

$$\gamma(P_1', P_2', \tau) = \frac{|\mu| |F(\tau)| e^{i[\varphi - 2\pi v_0 \tau + f(\tau)]}}{F_0}, \quad (7)$$

where $F(\tau)$ is the Fourier transform of $T(v)$ and $F(\tau) = |F(\tau)| e^{if(\tau)}$; $f(\tau)$ is the phase term associated with the shape of the spectrum. In particular, when $F(\tau)$ is symmetric about the central frequency $v_0$, $F(\tau)$ is a real function, $f(\tau) = 0$ or $\pi$. $F_0 = \int_0^\infty T(v) dv$, and $F_0$ can be considered as a constant.

Thus, the interference fringes generated at the coupler can be expressed as

$$I_{\text{tot}}(\tau) = I(P_1)' + I(P_2)' + \frac{2\sqrt{I(P_1)'} \sqrt{I(P_2)'} |\mu|}{F_0} |F(\tau)| \cos(\varphi - 2\pi v_0 \tau + f(\tau)), \quad (8)$$

where $I(P_1)'$ and $I(P_2)'$ represent the light intensity of the light signal at $P_1$ and $P_2$ after passing through the spectral channel of the imaging system, respectively.

## 2.2. Principle of Phase Difference Measurement

Note that $I_\Delta(\tau) = \frac{I_{\text{tot}}(\tau) - I(P_1)' - I(P_2)'}{2\sqrt{I(P_1)'}\sqrt{I(P_2)'}} = \frac{|\mu|}{F_0}|F(\tau)|\cos(\varphi - 2\pi v_0\tau + f(\tau))$ is the interference term, where $\frac{|\mu|}{F_0}$ is a constant, and let $\frac{|\mu|}{F_0}$=1. $|F(\tau)|$ can be regarded as the envelope function of the interference fringe. When $\tau = \frac{\varphi + 2n\pi + f(\tau)}{2\pi v}$, $I_\Delta(\tau)$ is a maximum value ($n$ is an integer), and when $\tau = \frac{\varphi + (2n+1)\pi + f(\tau)}{2\pi v}$, $I_\Delta(\tau)$ is a minimum value.

To simplify the analysis of the problem, assume that $T(v)$ is symmetric about the central frequency $v_0$, then $f(\tau) = 0$ or $\pi$. Further, assume that $f(\tau) = 0$ at two adjacent maximum positions of the interference fringe. For two interference signals with phases m1 and m2 ($0 \leq m1 \leq 2\pi$, $0 \leq m2 \leq 2\pi$), take the two adjacent maxima of $I_\Delta$ for the $\varphi = m1$ interference signal to be $a_n^{m1} = \left|F\left(\frac{m1 + 2n\pi}{2\pi v}\right)\right|$ and $a_{n+1}^{m1} = \left|F\left(\frac{m1 + 2(n+1)\pi}{2\pi v}\right)\right|$, and the first-order derivative $|F(\tau)|' = k$ is approximately constant when $\tau \in \left[\frac{m1 + 2n\pi}{2\pi v}, \frac{m1 + 2(n+1)\pi}{2\pi v}\right]$. The maximum of the interference signal of $\varphi = m2$ between $a_n^{m1}$ and $a_{n+1}^{m1}$ is $a_n^{m2} = \left|F\left(\frac{m2 + 2n\pi}{2\pi v}\right)\right|$. Then,

$$Rmax_n^m = \left|\frac{a_n^{m2} - a_n^{m1}}{a_n^{m1} - a_{n+1}^{m1}}\right| = \left|\frac{\left|F\left(\frac{m2 + 2n\pi}{2\pi v}\right)\right| - \left|F\left(\frac{m1 + 2n\pi}{2\pi v}\right)\right|}{\left|F\left(\frac{m1 + 2n\pi}{2\pi v}\right)\right| - \left|F\left(\frac{m1 + 2(n+1)\pi}{2\pi v}\right)\right|}\right|$$
$$\approx \left|\frac{k\frac{m2 - m1}{2\pi v}}{k\frac{2\pi}{2\pi v}}\right| = \frac{m2 - m1}{2\pi} = \frac{m}{2\pi} \quad , \tag{9}$$

where $m$ is the phase difference of the two interferometric signals. Similarly, $Rmin_n^m = \frac{m}{2\pi}$ can be calculated according to the minimum values of $I_\Delta$. Therefore, the phase difference of two interferometric signals with the phase difference $m$ satisfies $m = 2\pi Rmax_n^m = 2\pi Rmin_n^m$, i.e., the phase difference of two interferometric signals can be calculated based on the corresponding two adjacent maxima or minima of the interferometric signals. The interference term can be calculated from the interference fringe scanned by the integrated optical interferometric imager and the direct-current (DC) light intensity of the optical signal. After normalizing $I_\Delta$, a series of extrema of interference terms can be extracted, and the phase difference between different interference signals can be calculated.

## 2.3. Phase Difference Measurement Simulation

Assuming that $|\mu| = 1$, $T(v)$ is a ideal rectangular window function, i.e.,

$$T(v) = \begin{cases} 1, |v - v_0| \leq \frac{\Delta v}{2} \\ 0, |v - v_0| > \frac{\Delta v}{2} \end{cases} . \tag{10}$$

Then, $|F(\tau)| = \left|\frac{\sin(\pi\Delta v\tau)}{\pi\Delta v\tau}\right|$, $F(\tau)$ is a real function, $f(\tau)$= 0 or $\pi$, and $I_\Delta(\tau) = \frac{\sin(\pi\Delta v\tau)}{\pi\Delta v\tau}$ $\cos(2\pi v_0\tau - \varphi)$. Figure 2 shows the numerical calculation of the interference term $I_\Delta$ as well as $f(\tau)$ for an interferometric signal with a central wavelength of 1550 nm, a bandwidth of 25 nm, and $\varphi = 0$ in the OPD of $[-150\ \mu m, 150\ \mu m]$. It can be found that within the main lobe or each side lobe, the corresponding $f(\tau)$ of two adjacent extreme points is 0 or $\pi$.

When $\varphi = 0$, the series of maxima of the interference term normalized by the maximum value of its absolute value $|I_\Delta|_{max}$ are recorded as $a_1^0, a_2^0 \dots a_n^0$ ($n$ is a positive integer) in order of OPD; when $\varphi = m$, the maxima of the interference term are recorded as $a_1^m, a_2^m \dots a_n^m$ in order of OPD. A series of ratios $Rmax_n^m = \frac{(a_n^m - a_n^0)}{(a_{n+1}^0 - a_n^0)}$ can be calculated from the above extreme values. Figure 3 shows all the maxima $a_1^0, a_2^0 \dots a_n^0$ of the interference term extracted in the OPD $[-150\ \mu m, 150\ \mu m]$ at $\varphi = 0$ (i.e., the maxima extracted from Figure 2), and the absolute value of the difference between adjacent maxima $\Delta a_n^0 = |a_{n+1}^0 - a_n^0|$. The larger $\Delta a_n^0$ in the figure indicates the higher rate of change of $|F(\tau)|$. $\Delta a_n^0 = 0$ indicates that the first-order derivative of $|F(\tau)|$ is 0, while the intermittent points in the figure near $|F(\tau)| = 0$ indicate a break in $f(\tau)$, i.e., a jump in $f(\tau)$ from 0 to $\pi$, and $|F(\tau)|$ transitions between the two envelopes.

Figure 4 shows the individual $Rmax_n^m$ calculated from the extremum of interference terms with OPD of $[-150\ \mu m, 150\ \mu m]$ for $m = 0.2\pi, 0.4\pi, 1.0\pi, 1.4\pi, 1.8\pi$, respectively (the points of $Rmax_n^m > 1$ are omitted in the figure), and the results show that the value of $\left|Rmax_n^m - \frac{m}{2\pi}\right|$ is larger when the first-order derivative of $|F(\tau)|$ approaches 0 or when there is a jump in $f(\tau)$. Combining Figures 3 and 4, it can be seen that the relationship between the phase difference $m$ and the ratio $Rmax_n^m$ is most consistent with $m = 2\pi Rmax_n^m$ in the region where the second derivative of $|F(\tau)|$ in the main lobe or each side lobe is 0—that is, in the region where the envelope shape approximates a linear change—and the light intensity difference between adjacent extrema is the largest at this time. Therefore, the phase difference calculated from the extrema in the region where $|F(\tau)|$ varies approximately linearly is the most accurate.

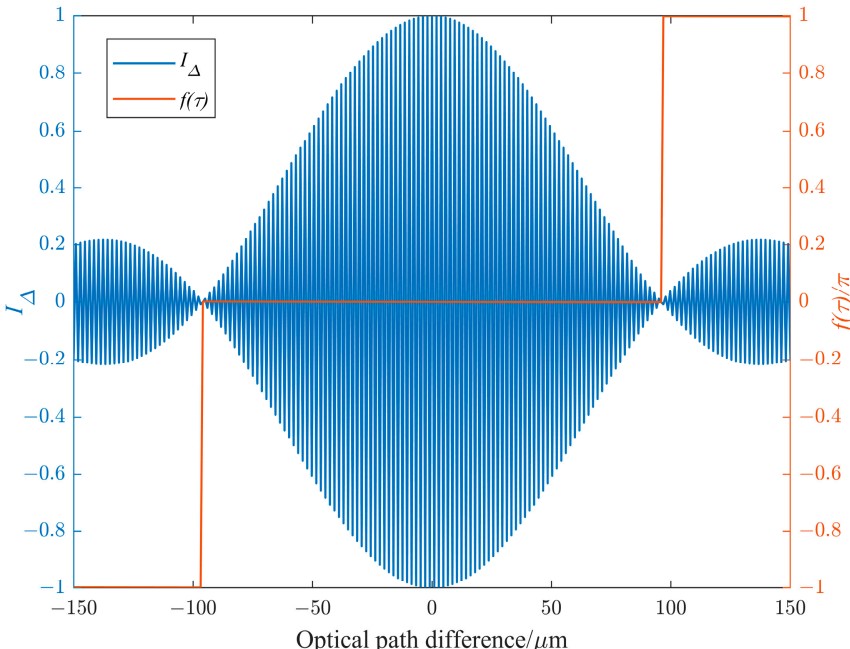

**Figure 2.** Simulated interference term $I_\Delta$ and phase $f(\tau)$ associated with the spectral shape for a range of optical path difference $[-150\ \mu m, 150\ \mu m]$.

For the same target, under the same experimental conditions of frequency response, dispersion, detector response, etc., the interference terms measured from different baselines have the same envelope shape. The difference is that the extrema will be enlarged or reduced by the same proportion when the modulus of the complex coherence factor is different; the extrema will be different when the phase of the complex coherence factor is different. Therefore, the phase difference between the interferometric signals corresponding to different baselines can be determined from the $Rmax_n^m$ or $Rmin_n^m$ of the normalized interference terms in the region of approximately linear variation of the envelope shape. From Figure 4, it can be seen that the difference between phase $m$ and $2\pi Rmax_n^m$ is less than $0.05\pi$, i.e., the phase difference measurement accuracy of the method is less than $0.05\pi$. Citing the simulation results of the influence of phase measurement error on reconstruction image quality in reference [36], when the mean value of the phase measurement error is 0, the standard deviation of the phase measurement error must be less than $0.18\pi$ to ensure the high quality of the reconstructed image. Therefore, the phase measurement accuracy of the proposed method is sufficient to ensure a high enough reconstruction image quality to meet the application requirements.

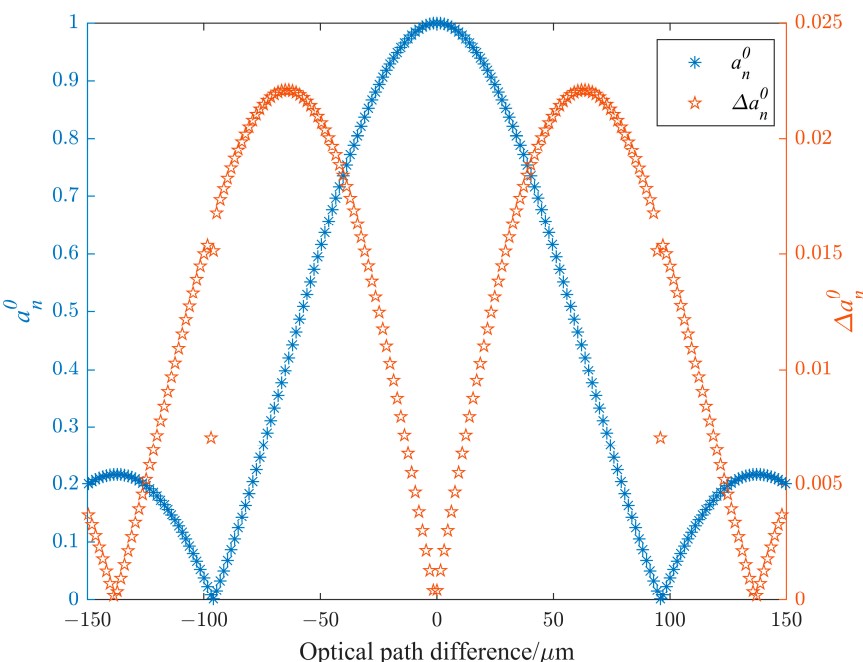

**Figure 3.** All maximum values $a_1^0, a_2^0 \ldots a_n^0$ of the interference term of $\varphi = 0$ in the range of optical path difference $[-150 \ \mu\text{m}, 150 \ \mu\text{m}]$, and the absolute value of the difference between adjacent maxima $\Delta a_n$.

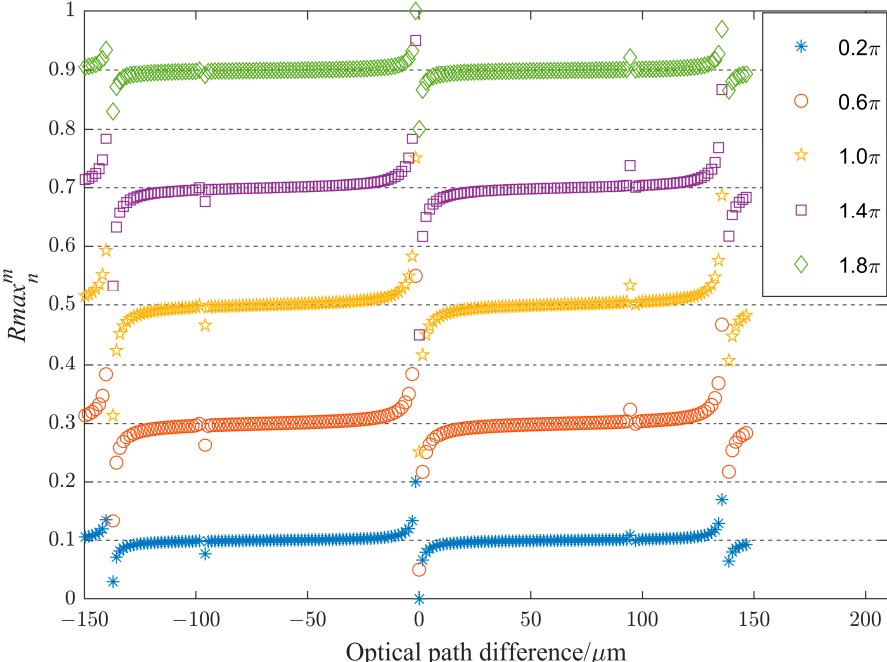

**Figure 4.** Calculated from all the maxima values $a_1^0, a_2^0 \ldots a_n^0$ of different phase interference signals in the range of optical range difference $[-150 \ \mu\text{m}, 150 \ \mu\text{m}]$.

## 3. Results

To verify the validity and feasibility of the method described above in calculating the phase difference, two optical test benches were built, and their experimental setups and experimental results are described below.

### 3.1. Amplitude-Division Interference Experiment

#### 3.1.1. Experimental Setup

Figure 5 shows the layout of the experimental setup using the amplitude-division interference scheme, which can be viewed as a Mach–Zehnder interferometer [37]. Figure 6 provides a photograph of part of the experimental setup. As shown in Figure 6a, a 150 W high-output halogen lamp (Thorlabs OSL2IR) was placed on the focal plane of an off-axis three-mirror collimator with 2.27 m focal length. The spectrum of the light source was modulated using a filter with a transmission spectrum as shown in Figure 7. As shown in Figure 6b, a fiber collimator (Thorlabs F280APC-1550) was used to couple the optical signal into the PM single-mode fibers. The $1 \times 2$ PM fiber coupler (splitting ratio 1:1, fast axis blocked) divided the optical signal into two paths—one was connected to the motorized fiber delay line and fiber stretcher, the other was only connected to the PM fiber, and the length of the two paths was similar. Finally, the $2 \times 2$ PM fiber coupler (splitting ratio 1:1, fast axis blocked) was used to combine the two signals, and a detector (Thorlabs PM101A, S154C) was used to record the intensity of the output optical signal. The motorized delay line with a delay range of up to 18 cm was used to compensate for a wide range of OPD, while the PM fiber stretcher with a delay range of 140 μm was used to scan interference fringes. The amplitude-division interference scheme can be viewed as the case where $P_1$ and $P_2$ overlap in Figure 1, i.e., the baseline length is 0. From Equations (4) and (5), we obtain $|\mu(P_1, P_2)| = \gamma(P_1, P_2, 0) = 1$, i.e., the theoretical maximum visibility of the interference fringe is 1.

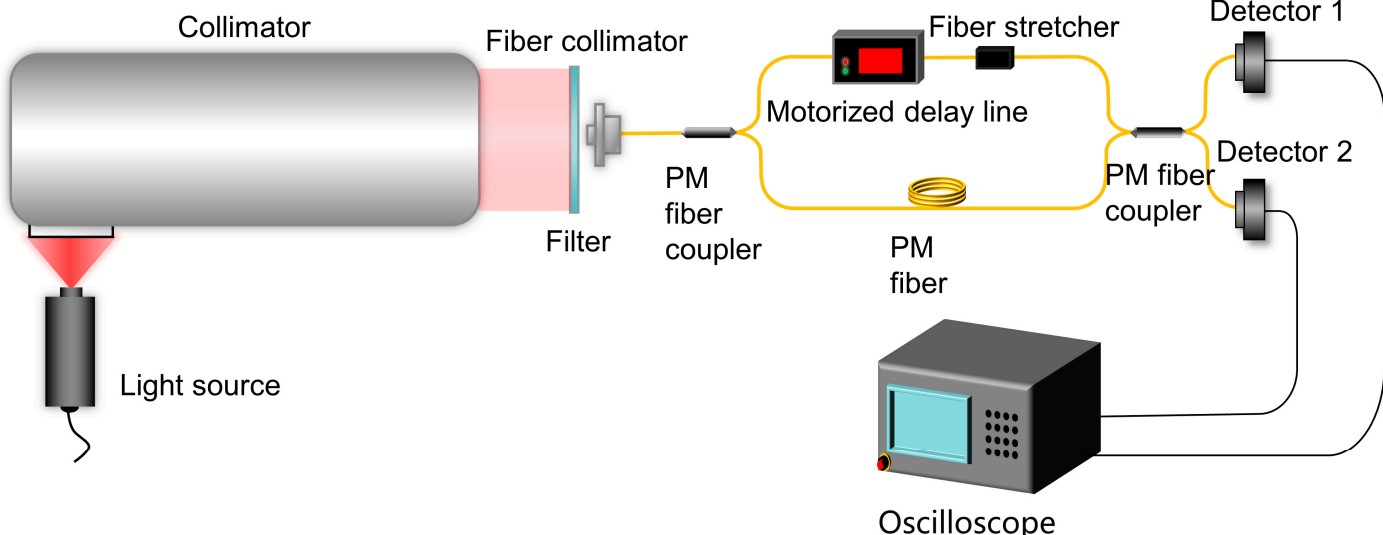

**Figure 5.** Layout of the experimental setup for amplitude-division interference.

During the output of the experiment, the analog signal from the detector is recorded with an oscilloscope (Tektronix MSO3034), and the effective value (root mean square, RMS) of the detector noise is measured to be 3.5 pW. After the interferometric signal to be measured with a phase $\varphi$ of the complex coherence factor passes through the $2 \times 2$ PM fiber coupler, there will be a phase shift of $\frac{\pi}{2}$ due to the mode coupling, resulting in the phase of the interferometric signals of the two outputs of coupler becoming $\varphi_+ = \varphi + \frac{\pi}{2}$ and $\varphi_- = \varphi - \frac{\pi}{2}$, respectively [38,39]. That is, the phase of the two outputs will differ by $\pi$. The validity and feasibility of the above method of measuring phase difference will be verified using these two interferometric signals with phase difference $\pi$.

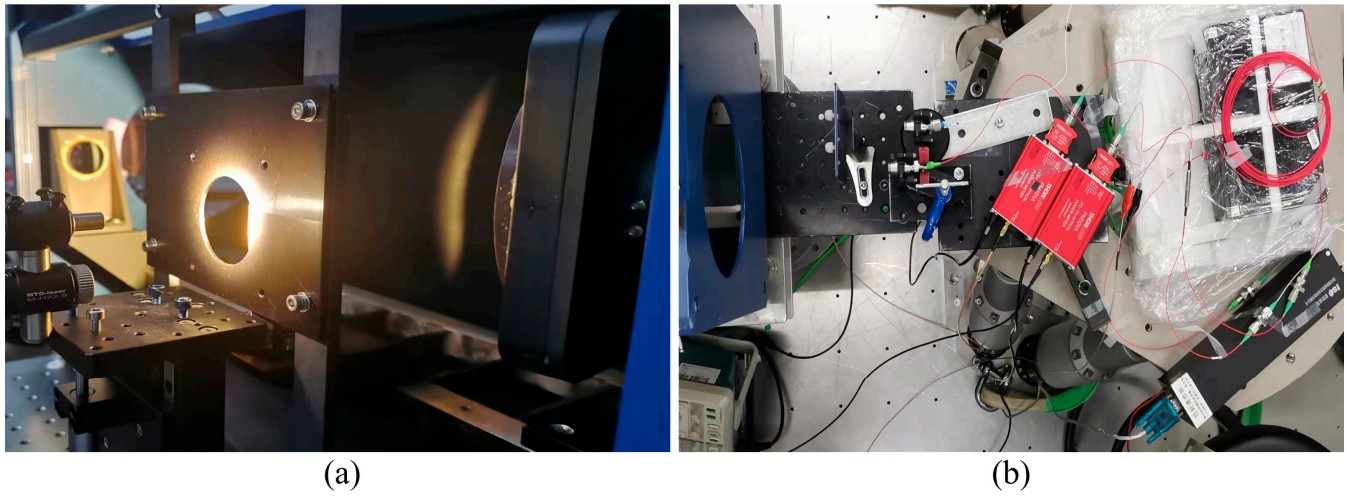

(a)          (b)

**Figure 6.** Photographs of the experimental setup: (**a**) light source, collimator with 2.27 m focal length; (**b**) filter, fiber patch cable, fiber stretcher, motorized delay line, coupler, detectors.

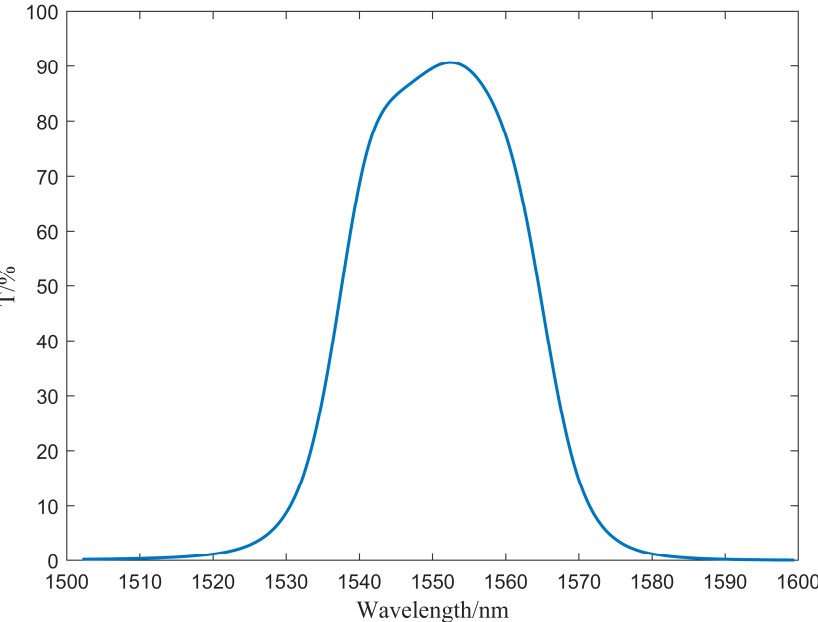

**Figure 7.** Transmission spectrum of the filter.

Figure 8 shows the interference term $I_\Delta$ scanned with the motorized delay line, and the appearance of multiple larger gaps in the figure is caused by the stagnation of the motor during the delay line operation. The blue and red curves in Figure 9 show the $I_\Delta$ calculated from the transmission spectrum and the phase $f(\tau)$ associated with the spectral shape, and it can be seen that the spacing between the two nearest zero points of the near zero OPD are spaced about 170 μm apart, and inside the main lobe, there is little variation at the two adjacent extrema. However, the OPD between the two zeros of the fringe envelope was found to be much larger than the coherence length corresponding to the transmission spectrum of the filter. This is because the experimental single-mode PM about 11 m in length introduces a certain degree of differential chromatic dispersion. Combined with Equation (7), the interference term $I_\Delta$ with dispersion is simulated after considering a certain dispersion in the phase term [40,41], as shown in the orange curve in Figure 9. It can be seen that the dispersion causes a significant change in the envelope of the interference term, the first zero point on both sides of the zero OPD disappears, and the maximum

visibility decreases. It is obvious that the simulated interference term after considering the dispersion is close to the measured interference term.

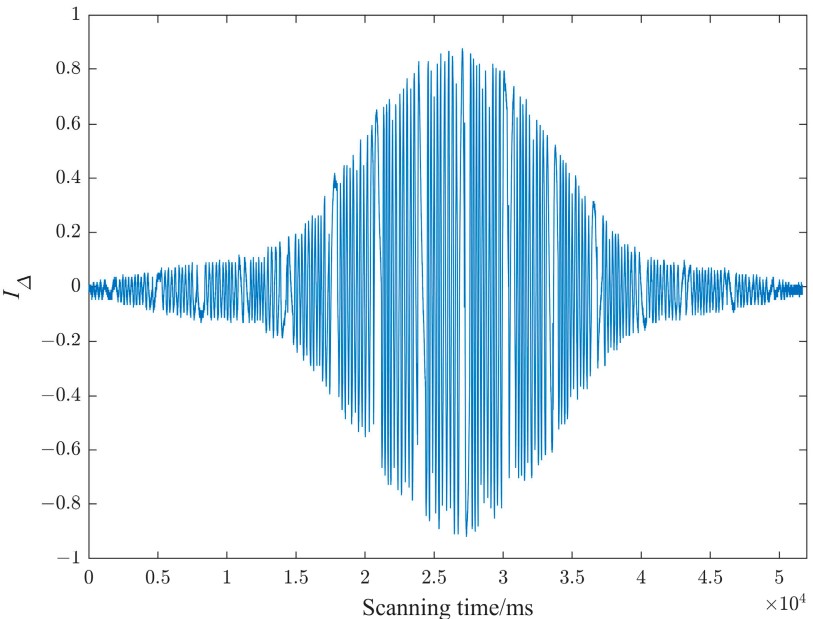

**Figure 8.** Measured interference term $I_\Delta$.

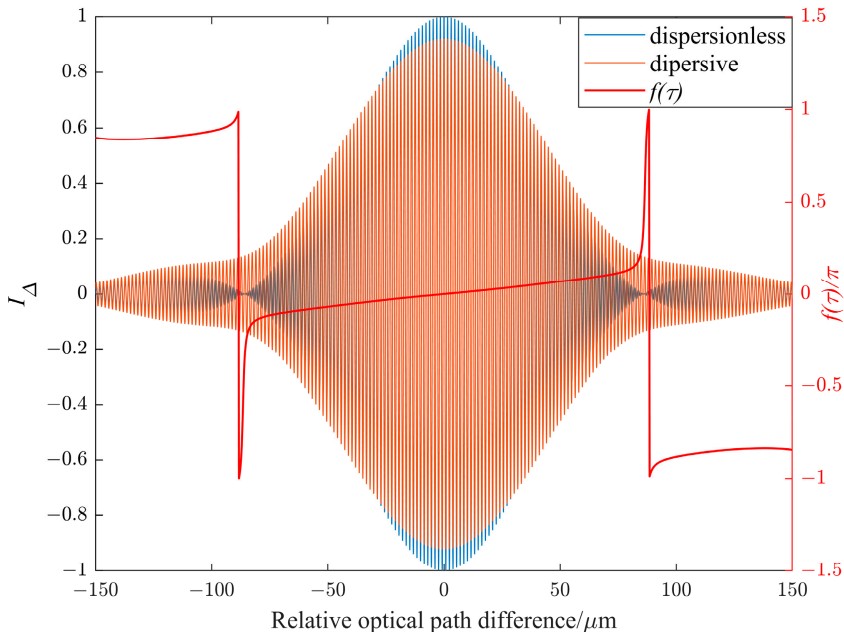

**Figure 9.** The simulated interference term $I_\Delta$, the phase $f(\tau)$ associated with the spectral shape in the absence of dispersion, and the interference term $I_\Delta$ in the presence of dispersion.

From the simulation analysis of Section 2.3, it can be seen that the phase difference between the two interferometric signals can be determined from the values of $Rmax_n^m$ and $Rmin_n^m$ calculated from the adjacent extrema in the region of approximately linear variation of the envelope shape. After normalizing the simulated interference terms in Figure 9, the interval in which the envelope shape varies approximately linearly is [0.5, 0.62]. Therefore, the phase difference of the two interference signals can be calculated using the extrema measured within [0.5, 0.62] of the normalized interference term in the experiment.

### 3.1.2. Experimental Results

Since the phase difference represented by $Rmax_n^m$ and $Rmin_n^m$ calculated at the extrema on both sides of the interferogram is the same, only one side of the interference fringe and the central region of the interference fringe need to be scanned using delay lines. Let the light intensities of the light signals involved in the interference be $I_1'$ and $I_2'$, and let the total light intensity of the interferometric signal be $I_{tot}'$. First, the raw data of the interferometric signal are preprocessed as follows: filtering, calculating the interference term $I_\Delta$, normalizing $I_\Delta$ using the absolute maximum value $|I_\Delta|_{max}$, obtaining the normalized interference term $\frac{I_\Delta}{|I_\Delta|_{max}}$, and extracting the extrema within [0.5, 0.62]. Figure 10 shows the original signal $I_{tot}$ acquired by the oscilloscope and the signal $I_\Delta$ filtered and subtracted from the DC light intensity.

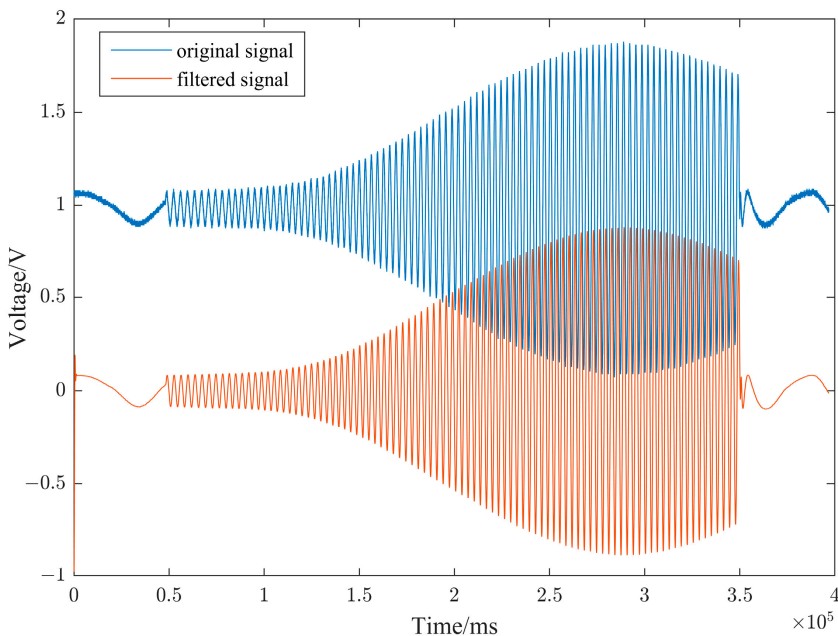

**Figure 10.** Original signal, and filtered signal with the direct-current light intensity subtracted.

The phase difference measurement method requires the measured extrema to be accurate enough, and the measurement results are affected by noise. To reduce accidental error during the extremum measurement, at least five measurements are made for each output, and the two outputs are measured asynchronously. In the experiment, the DC light intensity of each output of the $1 \times 2$ beam splitter in the two outputs of the $2 \times 2$ coupler is approximately equal. According to the simulation results, the difference between the adjacent maximum or minimum of the normalized interference term $\frac{I_\Delta}{|I_\Delta|_{max}}$ is about $a_n^0 = 0.023$, and the light intensity difference between the adjacent maximum or minimum points is $\Delta = 2\sqrt{I_1' I_2'} \Delta a_n^0 |I_\Delta|_{max}$. When the light intensity difference $\Delta$ between the extrema is greater relative to the detector noise, the measurement results are obviously more accurate. Let the signal-to-noise ratio (SNR) be $\Delta$ divided by the detector noise (3.5 pW). The experimental results of two groups with different SNRs are shown below. Table 1 shows the estimated SNR for the two sets of measured data, and $|I_\Delta|_{max}$ in the table does not reach 1, which is mainly caused by dispersion.

**Table 1.** SNR estimation for two sets of measured data.

| Data Type | $I'_1$ | $I'_2$ | $|I_\Delta|_{max}$ | $\Delta$ | SNR |
|---|---|---|---|---|---|
| $I'_1$ | | | | | |
| High SNR | 0.316 nW | 0.326 nW | 0.91 | 13.4 pW | 3.8 |
| Low SNR | 0.135 nW | 0.138 nW | 0.91 | 5.7 pW | 1.6 |

The results of the measurement at high SNR are shown in Figure 11. Figure 11a,b shows the absolute of normalized interference terms $\frac{I_\Delta}{|I_\Delta|_{max}}$ in the interval [0.5,0.62] for multiple measurements in the two outputs of the coupler, respectively, with the symbols $a$ for the extreme value, $b$ for the extreme small value, $n$ for the serial number of the extremum, and $m1$ and $m2$ for the different phases. It can be seen that the differentiation of different extrema is obvious. Figure 11c shows the results of averaging the measured normalized interference term $\frac{I_\Delta}{|I_\Delta|_{max}}$ at each corresponding location of the extrema, and Figure 11d shows the $Rmax_n^m$ and $Rmin_n^m$ calculated using these extrema. The mean value of all $Rmax_n^m$ is 0.54, the mean value of $Rmin_n^m$ is 0.52, and the standard deviation of all data is 0.034. According to $\varphi = \frac{Rmax_n^m + Rmin_n^m}{2} 2\pi$, the phase difference between these two outputs is calculated to be $1.06\pi$, which is $0.06\pi$ above the theoretical value.

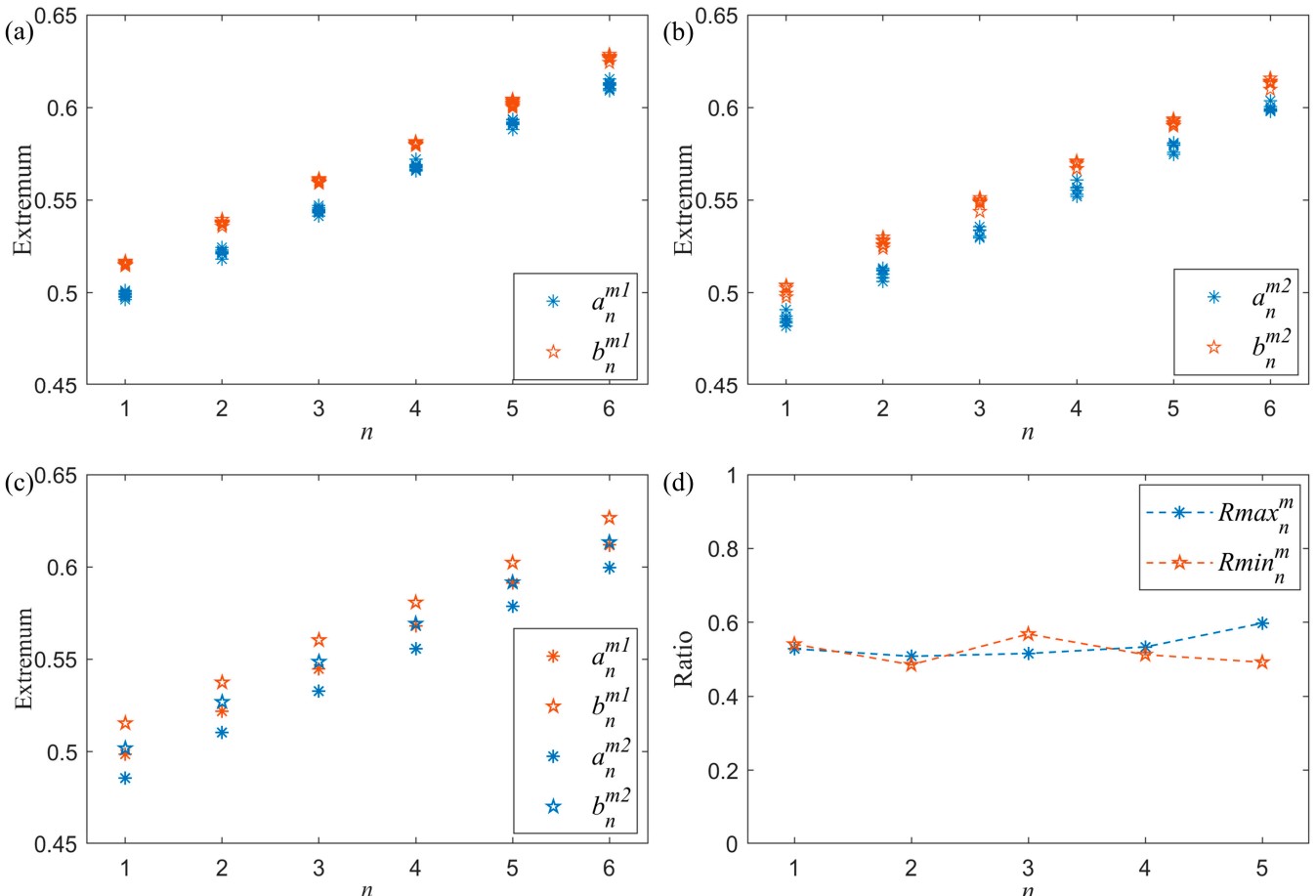

**Figure 11.** Measurement results of amplitude-division interference experiments at high signal-to-noise ratios. (**a**) extrema of the normalized interference term $\frac{I_\Delta}{|I_\Delta|_{max}}$ of the coupler output 1; (**b**) extrema of the normalized interference term $\frac{I_\Delta}{|I_\Delta|_{max}}$ of the coupler output 2; (**c**) average of the corresponding extrema of the normalized interference terms $I_\Delta$ of the two outputs; (**d**) $Rmax_n^m$ and $Rmin_n^m$ calculated from the extrema.

Figure 12 shows the measurement results at low SNR. Figure 12a,b shows the extrema of the normalized absolute interference terms $I_\Delta$ measured multiple times for the two outputs of the coupler, respectively, and it can be seen that there are still significant differences between the measured values at adjacent extrema, but the dispersion of all measured values at the same location of the extremum becomes larger compared to the data with high SNR. Figure 12c shows the average of all measurements of the corresponding extrema, and Figure 12d shows the $Rmax_n^m$ and $Rmin_n^m$ calculated using these extrema. The mean value of all $Rmax_n^m$ in the figure is 0.42, the mean value of $Rmin_n^m$ is 0.43, and the standard deviation of all ratios is 0.04, with an increase in the dispersion of the measured extrema. The phase difference between these two outputs is $0.85\pi$ according to $\varphi = \frac{Rmax_n^m + Rmin_n^m}{2} 2\pi$, which differs from the theoretical value by $0.15\pi$.

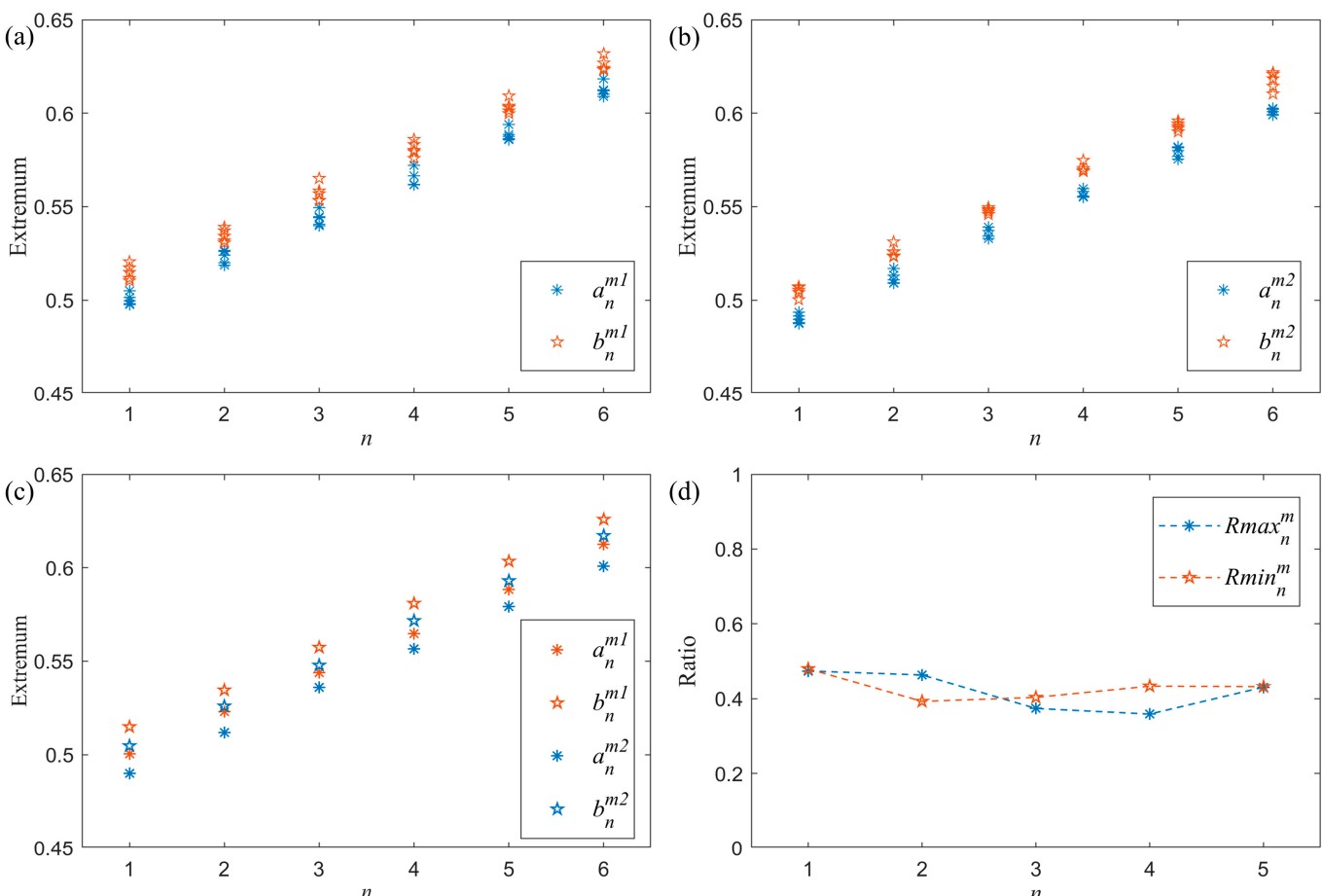

**Figure 12.** Measurement results of amplitude-division interference experiments at low signal-to-noise ratios. (**a**) extrema of the normalized interference term $\frac{I_\Delta}{|I_\Delta|_{max}}$ of the coupler output 1; (**b**) extrema of the normalized interference term $\frac{I_\Delta}{|I_\Delta|_{max}}$ of the coupler output 2; (**c**) average of the corresponding extrema of the normalized interference terms $\frac{I_\Delta}{|I_\Delta|_{max}}$ of the two outputs; (**d**) $Rmax_n^m$ and $Rmin_n^m$ calculated from the extrema.

### 3.2. Wavefront-Division Interference Experiment

3.2.1. Experimental Setup

Figure 13 shows the layout of the experimental setup with a wavefront-division interference scheme, which differs from the amplitude-division interference experiment in that a target is placed on the focal plane of a collimator and two fiber collimators are used to collect the optical signal. Figure 14 shows a centrosymmetric periodic grating target used in the experiment, with the black region being transmissive and the white region

being opaque, and each transmissive and opaque region being 50 µm wide and 2 mm long. This experimental setup layout is consistent with the principle of interferometric imaging, where the baseline consisting of two collimators produces interference fringes that contain both spatial and temporal coherence. If the complex coherence factor of a sufficient number of non-redundant baselines is measured, the target can be reconstructed.

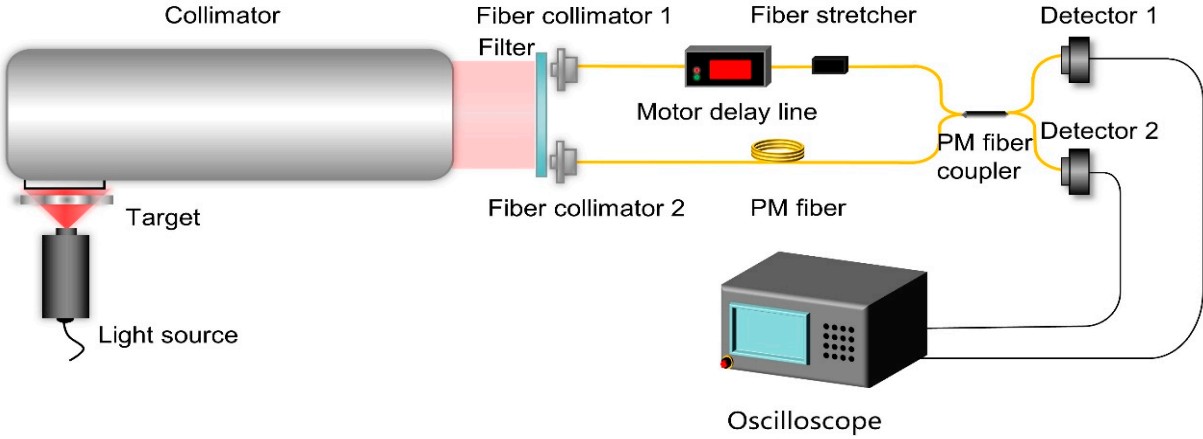

**Figure 13.** Layout of the experimental setup for wavefront-division interference.



**Figure 14.** A centrally symmetric periodic grating target.

3.2.2. Experimental Results

The baseline length was set to 3.5 cm, corresponding to the target eigenfrequency, where the measured fringe visibility was highest, i.e., the modulus of the complex coherence factor was maximum, thus maximizing the SNR. The DC output components of each collimator on the two outputs of the $2 \times 2$ coupler are approximately equal during the experiment. The following shows the experimental results for two sets of different SNRs. Table 2 shows the SNR estimates for the two sets of measurements.

**Table 2.** Estimation of the SNR for two sets of measured data.

| Data Type | $I_1'$ | $I_2'$ | $|I_\Delta|_{max}$ | $\Delta$ | SNR |
|-----------|--------|--------|--------------------|----------|-----|
| High SNR  | 0.276 nW | 0.316 nW | 0.46 | 6.2 pW | 1.8 |
| Low SNR   | 0.260 nW | 0.280 nW | 0.46 | 5.7 pW | 1.6 |

Figure 15 shows the results of the measurement at high SNR. Figure 15a,b shows the extrema of the normalized absolute interference term $\frac{I_\Delta}{|I_\Delta|_{max}}$ for the two outputs of the coupler, respectively, and it can be seen that there is some overlap in the distribution of multi-measurement results for adjacent extrema. Figure 15c shows the average of all measurements of the corresponding extrema. Figure 15d shows the $Rmax_n^m$ and $Rmin_n^m$

calculated using these extrema. The mean value of all $Rmax_n^m$ in the figure is 0.57, the mean value of $Rmin_n^m$ is 0.52, and the standard deviation of all ratios is 0.12, with an increase in the dispersion of the measured extrema. The phase difference between these two outputs is $1.09\pi$ based on $\varphi = \frac{Rmax_n^m + Rmin_n^m}{2}2\pi$, which differs from the theoretical value by $0.09\pi$.

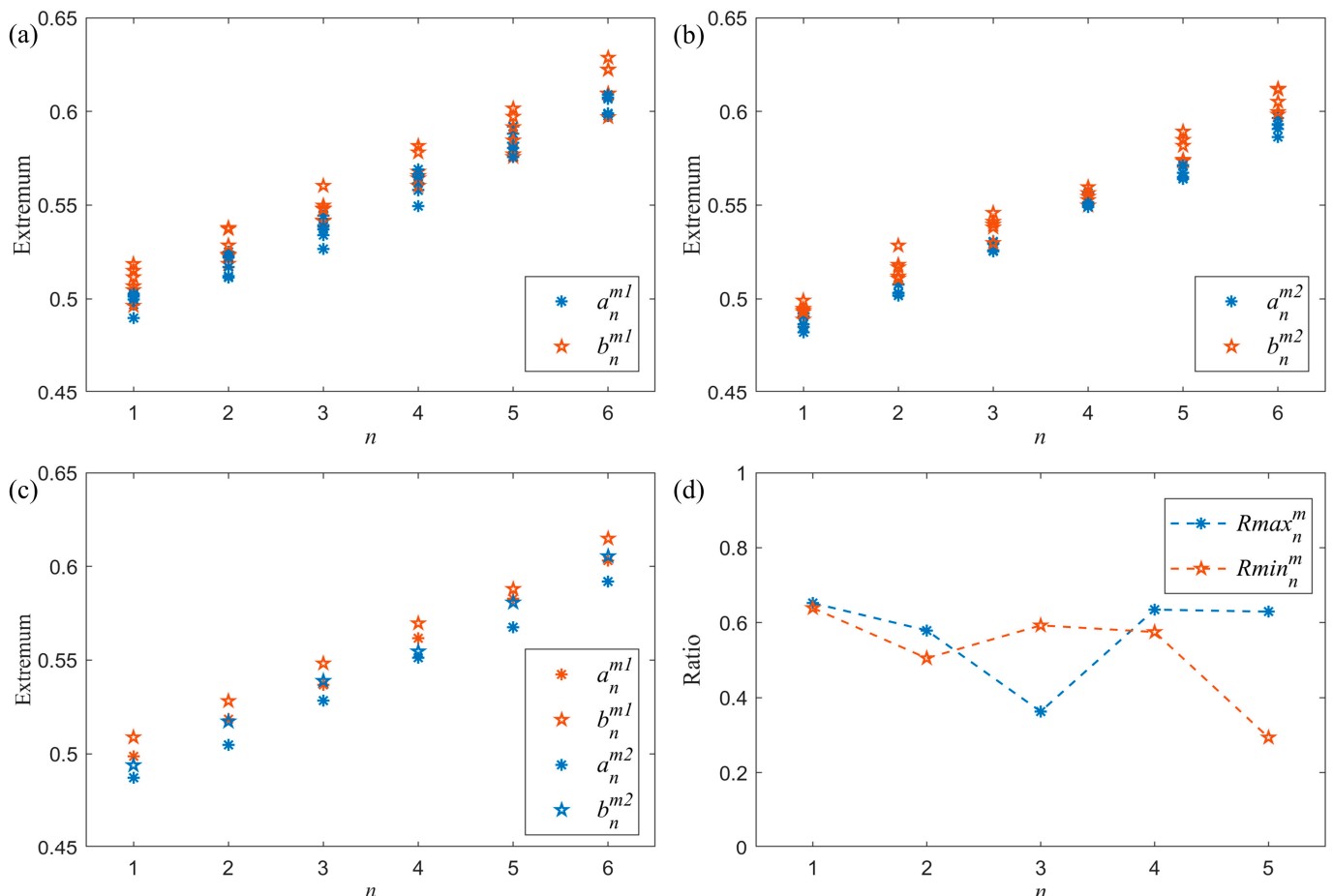

**Figure 15.** Measurement results of wavefront-division interference experiments at high signal-to-noise ratios. (**a**) Extrema of the normalized interference term $\frac{I_\Delta}{|I_\Delta|_{max}}$ of the coupler output 1; (**b**) extrema of the normalized interference term $\frac{I_\Delta}{|I_\Delta|_{max}}$ of the coupler output 2; (**c**) average of the corresponding extrema of the normalized interference terms $\frac{I_\Delta}{|I_\Delta|_{max}}$ of the two outputs; (**d**) $Rmax_n^m$ and $Rmin_n^m$ calculated from the extrema.

Figure 16 shows the measurement results at low SNR, based on the experimental conditions described above with the addition of a PM fiber patch cable of about 50 cm at the back end of the two collimators. Figure 16a,b shows the extrema of the normalized absolute interference term $\frac{I_\Delta}{|I_\Delta|_{max}}$ for the two outputs of the coupler, respectively, and Figure 16c shows the average of all the measurements of the corresponding extrema. Figure 16d shows the $Rmax_n^m$ and $Rmin_n^m$ calculated using these extrema. The mean value of all $Rmax_n^m$ in the graph is 0.36, the mean value of $Rmin_n^m$ is 0.49, and the standard deviation of all ratios is 0.13, which is a high degree of dispersion. The phase difference between these two outputs is calculated from $\varphi = \frac{Rmax_n^m + Rmin_n^m}{2}2\pi$ as $0.85\pi$, which is $0.15\pi$ below the theoretical value.

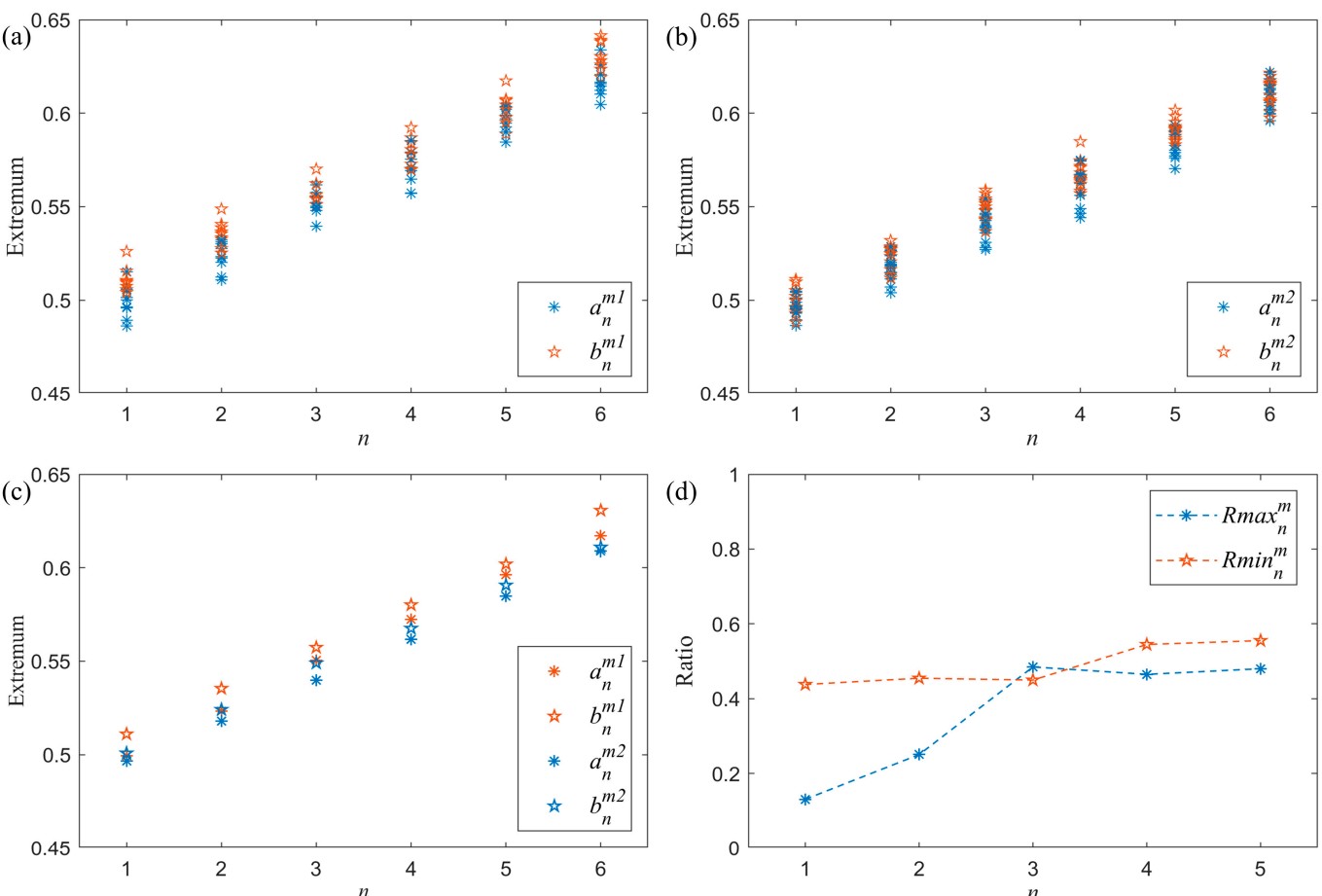

**Figure 16.** Measurement results of wavefront-division interference experiments at low signal-to-noise ratios. (**a**) extrema of the normalized interference term $\frac{I_\Delta}{|I_\Delta|_{max}}$ of the coupler output 1; (**b**) extrema of the normalized interference term $\frac{I_\Delta}{|I_\Delta|_{max}}$ of the coupler output 2; (**c**) average of the corresponding extrema of the normalized interference terms $\frac{I_\Delta}{|I_\Delta|_{max}}$ of the two outputs; (**d**) $Rmax_n^m$ and $Rmin_n^m$ calculated from the extrema.

## 4. Discussion

Combining the results of the above two experiments with a total of four sets of experimental data, it can be demonstrated that the phase difference between these two groups of signals can be calculated based on the extrema of the two interferometric signals in the region of approximately linear variation of the envelope shape. Due to the limitations of SNR, only the phase difference measurement results of two interferometric signals with phase difference $\pi$ are shown above, and the measurement error does not exceed $0.15\pi$. When comparing the above measurement results, it can be found that the measurement error of the phase difference is smaller when the SNR is higher. When comparing the measurements of the individual extrema, it can be found that when the SNR is the same, the extrema measured by the wavefront-division interference experiment are more discrete than those measured by the amplitude-division interference experiment. This may be due to the fact that the wavefront-division interference experiment contains two fiber collimators and is therefore more affected by the vibrations of the path from the light source to the fiber collimator (the two fiber collimators are placed on different platforms, and the light source will generate vibrations) [42]. It is also true that the drift of OPD is faster and wider in experiments of wavefront-division interference. In summary, the SNR of the optical intensity difference between the extrema, the drift of OPD, the vibration of the experimental environment, and the integration time of the detector all cause errors in the recorded inter-

ferometric signal extrema, thus leading to the measurement error of this phase difference measurement method. In addition, the results of the amplitude-division interferometric experiments illustrate that only one scan of the interference fringe is required to obtain sufficient information when the measurement error of the extrema is small.

Although the phase difference measurement method assumes that the frequency response of the imaging system is symmetric to the center frequency and the total dispersion of the transmission medium is zero, the experimental results show that the method is still applicable under experimental conditions where the frequency response is approximately centrosymmetric and the total dispersion of the transmission medium is small. The experiments in this paper are based on fiber devices, but the principle is consistent with the integrated optical interferometric imaging system. The measurement error caused by the OPD drift when scanning the interferometric signal based on the PIC will be smaller, and the dispersion corresponding to different baselines will also be smaller. For the same target, under the same conditions of frequency response, dispersion, detector response, etc., the interference fringes measured by the integrated optical interferometric imager at different baselines will have the same envelope shape, so the phase difference measurement method can be used to determine the phase difference between the interference signals corresponding to different baselines. As the SNR increases, the accuracy of the phase difference measurement method will be improved accordingly, and the phase difference of the interferometric signals with a small modulus of the complex coherence factor can be measured. The higher the accuracy of phase measurement, the higher the number of baselines that can be measured, and the richer the spatial spectrum information, which is conducive to reconstructing a better image.

## 5. Conclusions

In this paper, a method for calculating the phase difference of the complex coherence factor by comparing the extrema of the normalized interference terms of the interferometric signals in the region of approximately linear variation of the envelope shape is proposed. The validity and feasibility of the method were verified by two experimental schemes of amplitude-division interference and wavefront-division interference. The phase difference measurement method provides a new idea for measuring the phase of the complex coherence factor in integrated optical interferometric imaging because the integrated optical interferometric imager can more easily realize the scanning of the interferometric fringe. Compared with the existing phase measurement methods, the method does not require the calibration of zero OPD, which means that it can be applied to the integrated optical interferometric imager when a single-mode fiber is used to extend the baseline and can also make the imager work in a more flexible way, which can help to promote the application of the imager. The theoretical phase measurement accuracy of this method is higher than $0.05\pi$, which meets the image reconstruction requirements. Future work will focus on the implementation and application of this method in PIC.

**Author Contributions:** Conceptualization, J.C. and Q.Y.; methodology, J.C.; software, J.C. and B.G.; validation, J.C. and B.G.; formal analysis, J.C.; investigation, J.C.; resources, Q.Y.; data curation, J.C., B.G. and Q.Y.; writing—original draft preparation, J.C.; writing—review and editing, J.C., Q.Y., B.G., C.Z. and Y.H.; visualization, J.C., Q.Y. and Y.H.; supervision, Q.Y. and S.S.; project administration, Q.Y.; funding acquisition, Q.Y. All authors have read and agreed to the published version of the manuscript.

**Funding:** This study was funded by National Natural Science Foundation of China (No. 62105350) and the Youth Innovation Promotion Association of the Chinese Academy of Sciences (No. Y201951).

**Data Availability Statement:** The data presented in this study are available on request from the corresponding author.

**Conflicts of Interest:** The authors declare no conflict of interest.

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
