# Peer review of "A Phase Difference Measurement Method for Integrated Optical Interferometric Imagers"

_remotesensing, doi:10.3390/rs15082194_

Round 1
Reviewer 1 Report
Dear Author
I will present my comments in the order following the manuscript text.
The topic showed is from general interest and fits to the journal, the title is adequate, the content is well organized and the conclusions are according to the results presented.
1. Introduction
It is well written. It establishes the actual stage of the research that is exposed and the line that will be followed.
2. Body Text
The introduction is well written. It establishes the actual state of the research that is shown and the line of investigation that will be followed. The content and extension of the sections are suitable.
3. Conclusion
Conclusions are coherent with the results presented.
Mandatory review.
Add a photograph of your experimental arrangement with a corresponding description.
I recommended diversifying the references to improve the state of the art of the work presented and to establish the new contribution of his research.
B. Saif, P. Greenfield, M. North-Morris, M. Bluth, L. Feinberg, J. Wyant, R. Keski-Kuha, Sub-picometer dynamic measurements of a diffuse surface, App. Opt. 58(12) 3156-3165 (2019).
David-Ignacio Serrano-García, Amalia Martínez-García, Juan-Antonio Rayas-Alvarez, Noel I. Toto Arellano Sr., Gustavo Rodriguez-Zurita, and Areli Montes Pérez "Adjustable-window grating interferometer based on a Mach-Zehnder configuration for phase profile measurements of transparent samples," Optical Engineering 51(5), 055601 (4 May 2012)
Virendra N. Mahajan; Daewook Kim, Tribute to James C. Wyant: The Extraordinaire in Optical Metrology and Optics Education, Proc. SPIE 11813 (2021)
Author Response
Response to the comments of Reviewer 1
Comment No.1: Add a photograph of your experimental arrangement with a corresponding description.
Response: We have added photos of the experimental setup, as shown in Figure 6 of the revised manuscript, with the corresponding description in the first paragraph of Sect. 3.1.1.
Comment No.2: I recommended diversifying the references to improve the state of the art of the work presented and to establish the new contribution of his research.
Response: We thank the reviewer for his suggestion. The literature recommended by the reviewer has broadened our horizons, and we have read and cited them in the revised manuscript.
Reviewer 2 Report
The manuscript is well-written, clear, precise, and easy to understand. The methodology is satisfactory, and the results are impressive.
The overall level of the paper is good. Based on these, the paper has the potential to be accepted, but minor revisions are required to make the manuscript worth publishing which is mentioned below.
1. Clear statements of the novelty of the work should also appear briefly in the Abstract and Conclusions sections.
2. An updated and complete literature review should be conducted and should appear as part of the Introduction, while bearing in mind the work's relevance to this Journal and taking into account the scope and readership of the journal. The results and findings should be compared to and discussed in the context of earlier work in the literature.
3. Manuscript needs to improve the English, typographical errors, and end nodes.
4. In the reference section, some information's missing.
5. The authors should mention which are the motivation papers!
6. I found that the Preliminaries section is weak; the authors may include the necessary basic results, which will be good for readers.
Author Response
Response to the comments of Reviewer 2
Comment No.1: Clear statements of the novelty of the work should also appear briefly in the Abstract and Conclusions sections.
Response: Thanks to the reviewer's suggestion. The novelty of the work in this paper is mainly in proposing a method for calculating the phase difference of the complex coherence factor using the light intensity extremums of the interferometric fringe. This method does not need to calibrate the position of the zero optical path difference, and can be applied to the integrated optical interferometric imager using a single-mode fiber, which also allows the imager to work in a more flexible way. The theoretical phase measurement accuracy of this method is higher than 0.05 , which meets the image reconstruction requirements. We have made relevant amendments to the Abstract and Conclusion sections.
Comment No.2: An updated and complete literature review should be conducted and should appear as part of the Introduction, while bearing in mind the work's relevance to this Journal and taking into account the scope and readership of the journal. The results and findings should be compared to and discussed in the context of earlier work in the literature.
Response: We thank the reviewers for pointing out the problem in the Introduction. The integrated optical interferometric imager is a newly proposed imaging system that is suitable as a payload for planetary detection in space because of its greatly reduced mass, size, and power consumption compared to conventional telescope systems at the same imaging resolution. This paper addresses the key technologies of such remote sensing payloads and thus fits well with the theme of this special issue: "Laser and Optical Remote Sensing for Planetary Exploration". An update and complete literature review on the imager has been added in the Introduction, and relevant references are added to give the readers a better understanding of this imager.
As for the lack of direct comparison and discussion of the results and findings with earlier work in the literature, the explanation is as follows. In this paper, the following three methods for measuring the phase of the complex coherence factor are introduced in the Introduction section, determining the position of the zero optical path difference (OPD) through the channel spectrum, determining the position of the zero OPD through the envelope fitting method, and advance calibration of the zero OPD position. In the Introduction, a qualitative comparison of the differences and advantages and disadvantages of the three methods has been made. Our paper focuses more on the complex coherence factor phase measurement method for integrated optical interferometric imagers. Whereas the channel spectrum method typically used for large optical interferometers is more suitable for broadband spectroscopy and it increases the difficulty of integration when used for integrated optical interferometric imagers, the results and findings of this paper are not suitable for comparison with earlier work on channel spectrum methods. Regarding the envelope fitting method, although the method can be used to determine the envelope position and the approximate center of the envelope, it cannot accurately fit the zero OPD position, and there are no published experimental results of direct phase measurements using the method. The method of pre-calibrating the zero OPD position is only demonstrated in the following work: Badham, K.; Kendrick, R.; Wuchenich, D.; et al. Photonic integrated circuit-based imaging system for SPIDER. In Proceedings of the Conference on Lasers and Electro-Optics Pacific Rim (CLEO-PR), 2017. doi:https://doi.org/10.1109/CLEOPR.2017.8118616), but the details of its measurement results are not shown in that literature. Therefore, the results and findings in this paper are not compared with earlier work in the literature. For further comparison and discussion of the experimental results and findings with existing methods in terms of accuracy, advantages, and performance in practical engineering applications, see also the responses to the first two questions of Reviewer 3.
Comment No.3: Manuscript needs to improve the English, typographical errors, and end nodes.
Response: Thanks a lot for reviewer’s kind reminding and our manuscript has been polished by a paid editing service.
Comment No.4: In the reference section, some information's missing.
Response: Thanks for the reviewer’s reminding, we have added the following information to the references.
- Added publisher information to references 2 and 14 of the pre-revision manuscript;
- Added patent version number and date to reference 11 of the pre-revision manuscript;
3.Added the digital object identifier (DOI) to references 9 of the pre-revision manuscript.
Comment No.5: The authors should mention which are the motivation papers!
Response: We have supplemented the research status of integrated optical interferometric imagers in the Introduction and described the motivation of this article at the second paragraph of Introduction.
Integrated optical interferometric imagers have promising applications in the field of space target detection. The motivation for this paper stems from the difficulty in measuring the phase of the complex coherence factor when the fiber is introduced to enhance the resolution of the imager by extending the baseline through the fiber. This paper is based on a thorough observation of the experimental phenomena and an in-depth consideration of the physical principles of the phase of the complex coherence factor, which leads to the conclusion of a phase difference measurement method. The related literature on the phase of the complex coherence factor has also been cited in the beginning of Materials and Methods.
Comment No.6: I found that the Preliminaries section is weak; the authors may include the necessary basic results, which will be good for readers.
Response: We have supplemented the current research status of integrated optical interferometric imagers in the Introduction; and made modifications to the first and third paragraphs of the Introduction of the revised manuscript to help readers better understand the research content of this article.
Reviewer 3 Report
A method to calculate the phase difference of the complex coherence factor of two interference signals is proposed in the manuscript “A phase difference measurement method for integrated optical interferometric imagers”. In my opinion, the authors still need to make some necessary modifications to improve the quality of the paper. My questions or suggestions are as follows:
(1) Compared with existing methods, whether the proposed method has significant advantages in efficiency or accuracy. If the proposed method cannot meet or exceed the actual detection accuracy of existing methods, then even without pre-calibration requirement, in my opinion, the application prospects are also limited. It is recommended to include a comparison with other methods in the manuscript to highlight the advantages.
(2) The performance in practical engineering applications is an indicator for judging the merits of the proposed method, whether the proposed method meets the requirements of current applications.
(3) It is noticed that there are two obvious jump positions for the change of the value of Δa0n.In Figure 3, what is the reason to explain the phenomenon.
(4) The presentation is still redundant in some sections, such as in the conclusion section, where the author should summarize the work content concisely and highlight the core content.
(5) The manuscript is required to be polished to enhance the expression.
Author Response
Response to the comments of Reviewer 3
Comment No.1: Compared with existing methods, whether the proposed method has significant advantages in efficiency or accuracy. If the proposed method cannot meet or exceed the actual detection accuracy of existing methods, then even without pre-calibration requirement, in my opinion, the application prospects are also limited. It is recommended to include a comparison with other methods in the manuscript to highlight the advantages.
Response: There are three main methods for measuring the phase of the complex coherence factor as follows: determining the position of the zero optical path difference (OPD) through the channel spectrum, determining the position of the zero OPD through the envelope fitting method, and advance calibration of the zero OPD position. In the Introduction, a qualitative comparison of the differences and advantages and disadvantages of the three methods has been made.
The channeled spectrum method is more suitable for wider bandwidth and will increase the difficulty of integration for integrated optical interferometric imagers. Although the envelope fitting method can be used to determine the envelope position and the approximate center of the envelope, it cannot precisely fit the zero OPD position, and no experimental results have been published that directly use this method to measure the phase. In fact, at the beginning, this paper also tried to use the envelope fitting method to locate the position of zero OPD, but the results were not satisfactory, so this paper is looking for other methods to measure the phase. The method of pre-calibrating the zero OPD position is only demonstrated in the following work: Badham, K.; Kendrick, R.; Wuchenich, D.; et al. Photonic integrated circuit-based imaging system for SPIDER. In Proceedings of the Conference on Lasers and Electro-Optics Pacific Rim (CLEO-PR), 2017. doi:https://doi.org/10.1109/CLEOPR.2017.8118616), but the details related to the efficiency and accuracy of its phase measurement are not shown in this work. Therefore, in this paper, no quantitative comparison of the efficiency and accuracy of different methods with respect to the experimental results is presented.
The most obvious advantage of the proposed method in this paper is that it does not need to calibrate the zero OPD position and thus can measure the phase in the case of OPD change, such as using fiber to extend the baseline to improve the resolution of the imager or changing the orientation of the imager, and other cases. Therefore, the method proposed in this paper can make the working mode of the integrated optical interferometric imager more flexible and help promote the application of the imager.
Further analysis of the accuracy or efficiency of the proposed method can be found in the response to Comment No. 2. Again, we are grateful to the reviewer for this recommendation and we have added the accuracy and advantages of the proposed method in the Abstract and Conclusion sections of the revised manuscript.
Comment No.2: The performance in practical engineering applications is an indicator for judging the merits of the proposed method, whether the proposed method meets the requirements of current applications.
Response: The key for the method to meet the application requirements is whether its phase difference measurement accuracy meets the reconstruction image requirements. The simulation results of Figure 4 in the manuscript show that the phase difference measurement accuracy of the proposed method is less than 0.05 . Citing the simulation results of the effect of phase measurement error on reconstruction image quality in the following work: Chen J, Ge B, Yu Q. Influence of measurement errors of the complex coherence factor on reconstructed image quality of integrated optical interferometric imagers. Optical Engineering, 2022, 61(10): 105108. When the mean value of the phase measurement error is 0, the standard deviation of the phase measurement error must be less than 0.18π to ensure the high quality of the reconstructed image. Therefore, the phase measurement accuracy of the proposed method is sufficient to ensure a high enough reconstruction image quality to meet the application requirements. This analysis has been added at the end of Section 2.3 of the manuscript.
To meet the measurement accuracy achieved by theory in practical application, it has been mentioned in the paper that the measurement accuracy is affected by the signal-to-noise ratio, the drift of the OPD, the vibration of the experimental environment, the integration time of the detector, and other factors. Therefore, future studies will continue to improve and optimize these factors, especially in selecting the appropriate detector.
The hardware basis for the application of this method is the implementation of delay lines and phase shifters in the photonic integrated circuits and the automation of data processing. The response speed of current on-chip phase shifters is sufficient to meet demand, but it may be difficult to achieve a wide range of phase delays. Fortunately, the method proposed in this paper only needs to accurately measure the light intensity at a few extreme points, and the requirement for scanning range is greatly reduced. Therefore, the method proposed in this paper is definitely achievable with the current device performance level. The hardware implementation and data processing automation of the proposed method will also be the focus of subsequent research in this paper.
Comment No.3: It is noticed that there are two obvious jump positions for the change of the value of Δa0n. In Figure 3, what is the reason to explain the phenomenon.
Response: The phenomenon has been described in the manuscript. As in line 193 of Sec. 2.3 of the pre-revision manuscript, "while the intermittent points in the figure near =0 indicates that appears to step, i.e., transitions between the two envelopes". The intermittent points are the two obvious jump positions, in which (the phase of the complex function , here ) will change from 0 to . When is a real function, will change from positive to negative, and the difference between the extremes of adjacent can be seen as continuous, but will produce jumps, and thus will also produce jumps. " a jump in from 0 to " has been added to the manuscript to make the expression clearer.
Comment No.4: The presentation is still redundant in some sections, such as in the conclusion section, where the author should summarize the work content concisely and highlight the core content.
Response: Thanks to the reviewers' suggestions, we have revised the Discussion and Conclusion sections to make the Conclusion section more concise and to highlight the significance of the research work in this paper. The significance of the research work in this paper is mainly in proposing a method for calculating the phase difference of the complex coherence factor using the light intensity extremums of the interferometric fringe. This method does not need to calibrate the position of the zero optical path difference, and can be applied to the integrated optical interferometric imager using a single-mode fiber, which also allows the imager to work in a more flexible way. The theoretical phase measurement accuracy of this method is higher than 0.05 , which meets the image reconstruction requirements.
Comment No.5: The manuscript is required to be polished to enhance the expression.
Response: Thanks a lot for the kind reminder of the reviewer. Our manuscript has been polished by a paid editing service.
Round 2
Reviewer 2 Report
The revised manuscript is well-written, clear, precise, and easy to understand. The methodology is satisfactory, and the results are impressive.
The overall level of the paper is good. Based on these, the paper has the potential to be accepted, but minor revisions are required to make the manuscript worth publishing, which is mentioned below.
1. Clear statements of the novelty of the work should also appear briefly in the Abstract and Conclusions sections.
2. An updated and complete literature review should be conducted and appear as part of the Introduction while considering the work's relevance to this Journal and considering the journal's scope and readership. The results and findings should be compared to and discussed in the context of earlier work in the literature.
3. Manuscript needs to improve the English, typographical errors, and end nodes.
4. In the reference section, some information's missing.
5. The authors should mention more motivation papers in this field!
6. I found that the Preliminaries section is weak; the authors may include the necessary primary results, which will be good for readers.
Reviewer 3 Report
The reviewer's concern was well addressed.
Author Response
Thank you for your recognition.